

# Impacts of temperature extremes on European vegetation during the growing season

Lukas Baumbach[1,2], Jonatan F. Siegmund[1,3], Magdalena Mittermeier[1,4], and Reik V. Donner[1]

[1]Research Domain IV – Transdisciplinary Concepts and Methods, Potsdam Institute for Climate Impact Research, Telegrafenberg A31, 14473 Potsdam, Germany

[2]Faculty of Environment and Natural Resources, Albert-Ludwigs-University Freiburg, Tennenbacherstraße 4, 79016 Freiburg im Breisgau, Germany

[3]Institute of Earth and Environmental Science, University of Potsdam, Karl-Liebknecht-Straße 24-25, 14476 Potsdam, Germany

[4]Department of Geography, Ludwig Maximilians University, Geschwister-Scholl-Platz 1, 80539 München, Germany

*Correspondence to:* Reik V. Donner (reik.donner@pik-potsdam.de)

**Abstract.** Temperature is a key factor controlling plant growth and vitality in the temperate climates of the mid-latitudes like in vast parts of the European continent. Beyond the effect of average conditions, the timings and magnitudes of temperature extremes play a particularly crucial role, which needs to be better understood in the context of projected future rises in the frequency and/or intensity of such events. In this work, we employ event coincidence analysis (ECA) to quantify the likelihood

of simultaneous occurrences of extremes in daytime land surface temperature anomalies and the normalized difference vegetation index (NDVI). We perform this analysis for entire Europe based upon remote sensing data, differentiating between three periods corresponding to different stages of plant development during the growing season. In addition, we analyze the typical elevation and land cover type of the regions showing significantly large event coincidences rates to identify the most severely affected vegetation types. Our results reveal distinct spatio-temporal impact patterns in terms of extraordinarily large co-

occurrence rates between several combinations of temperature and NDVI extremes. Croplands are among the most frequently affected land cover types, while elevation is found to have only a minor effect on the spatial distribution of corresponding extreme weather impacts. These findings provide important insights into the vulnerability of European terrestrial ecosystems to extreme temperature events and demonstrate how event-based statistics like ECA can provide a valuable perspective on environmental nexuses.

# 1 Introduction

At a global scale, the year 2016 has been the warmest year on record, presenting the third annual record among more than 100 years of observational data (NOAA National Centers for Environmental Information, 2017). For Europe, it was still the 3rd warmest year (after 2014 and 2015). In the context of ongoing anthropogenic greenhouse gas emissions, these general tendencies are likely to continue, resulting in an increasing trend of extreme temperature events (Beniston et al., 2007; Coumou

et al., 2013). Particularly hot days and heat waves have been observed to increase in intensity and frequency in the course of the last few decades, where prominent examples include the European summer heat waves of 2003, 2006 and 2010 (De Bono




et al., 2004; Rebetez et al., 2009; Barriopedro et al., 2011). Additionally, late spring frost events may pose a threat to plant development due to continuously earlier bud burst of plants in spring (Way, 2011).

Both positive and negative temperature extremes can adversely affect a plant's development and vitality. On the one hand, heat stress leads to changes in plant metabolism and the integrity of cells, causing phenomena such as the inhibition of pho-
tosystem II, thermal denaturation of proteins or the formation of reactive oxygen species (Hasanuzzaman et al., 2013). On the other hand, low temperatures and frosts slow down biochemical processes within the plant cells and may result in the direct damaging or necrosis of new shoots or plant roots as well as indirect damages due to the lack of water access in the frozen soil (frost drought) (Beck et al., 2004). The susceptibility of vegetation to these threats may vary, e.g., depending on the seasonal stage of development (*phenophase*) and vegetation type (see Section 3). Foremost, cultivated plants/croplands show high vul-
nerabilities to extreme temperatures as their optimal climatic niche is relatively narrow (Larcher, 1994). Thus, temperature as a single variable can already have severe impacts and even be of predominant importance for future agricultural production in Europe (Semenov and Shewry, 2011; Deryng et al., 2014). At the prospect of increasing frequencies and/or intensities of extreme temperature events in Europe (Coumou and Rahmstorf, 2012; Tank and Konnen, 2003; Luterbacher et al., 2004; IPCC, 2013; Barriopedro et al., 2011; Petoukhov et al., 2013; Seneviratne et al., 2012), plant communities will in general face ever
more challenges to resist the stress imposed on them.

A widely used measure for identifying plant stress at regional scales based on remote sensing observations is the Normalized Difference Vegetation Index (NDVI). While other more recently developed indices for vegetation characteristics (such as the Enhanced Vegetation Index or the Net Primary Productivity) have shown to yield advantages over the NDVI in terms of stress detection in plants, the NDVI is still in frequent use. Despite its lower sensitivity to changes of the vegetation cover, the NDVI
has proven a conservative, robust measurement tool (Pettorelli, 2013) and has been comprehensively studied regarding its relationship with temperature variability (see, e.g., Schultz and Halpert, 1993, 1995; Kawabata et al., 2001; Karnieli et al., 2010; Kim et al., 2010). The global data availability for a relatively long period of observations has made NDVI easily applicable to remote sensing studies without the need for exhaustive preprocessing. Because of this, we focus in this work on the analysis of NDVI characteristics while outlining refined studies using more sophisticated indices as a subject of future research.

Up to now, research on climatic drivers of NDVI variations has often concentrated on a global scale (e.g. the aforementioned Schultz and Halpert, 1993, 1995; Kawabata et al., 2001; Karnieli et al., 2010; Kim et al., 2010), whereas regional studies like Wang et al. (2003) and Hao et al. (2012) are still rare. In the face of global warming, however, especially inter-regional climate disparities are projected to increase (Ciscar et al., 2011; Iglesias et al., 2012), thus emphasizing the need for regional research.

Moreover, most previous studies on the environmental impacts of changing temperatures have focused on the detection
of linear relationships between different variables by means of correlation and regression analysis (see, e.g., Schultz and Halpert, 1995; Los et al., 2001; Ichii et al., 2002; Wang et al., 2003; Stöckli and Vidale, 2004; Hao et al., 2012). Linear correlation analysis is a widely applicable tool allowing to discover linear statistical relationships between environmental variables. However, corresponding changes associated with global warming cannot a priori be assumed to be of linear nature, but may, for example, only occur when a certain threshold is exceeded (see, e.g., Burkett et al., 2005; Rockström et al., 2009).
Liu et al. (2013) took this idea a step further by investigating the sensitivity of vegetation to climate extremes by conducting



a pixel-wise regression between extremes in vegetation dynamics subject to a Box-Cox data transformation and extremes of precipitation, Palmer Drought Severity Index and temperature.

While the aforementioned approach explored on anomalies in the data distribution and took into account a broad ensemble of drought-related indicators, in this work, we specifically focus on analyzing the impacts of extraordinary warm or cold

temperature events by utilizing the novel and straightforward method of *event coincidence analysis* (ECA, Donges et al., 2016). This method differs from linear correlation analysis in that it does not assess the dependency between two variables as a whole, but focuses on simultaneous occurrences of specific events. The theoretical differences between correlation analysis and ECA, as well as the differences in the interpretations of the outcomes of both methods have been comprehensively discussed in previous papers (Donges et al., 2016; Siegmund et al., 2016a, b, 2017). In summary, correlation analysis reveals the general

common behavior (covariance) between two time series, while ECA addresses the commonality in the timing of occurrences of values in a specific part of the empirical distributions (in our case, the tails) of the variables of interest.

Drawing upon the previous considerations, this study aims to identify regions of Europe with a vegetation cover that is particularly sensitive to temperature extremes. For this purpose, we utilize the land surface temperature anomalies during daytime (LSTAD) as a temperature variable and the NDVI as a proxy for the vegetation condition. By studying the behavior

during three distinct phenophases individually, we will distinguish between different types of temperature-vegetation relations during different parts of the growing season. For each of these phases, we additionally study the distributions of statistically significant event coincidence rates among different classes of land cover and elevation to unveil which types of terrestrial ecosystems may be affected the most by extremal temperature conditions.

The remainder of this paper is organized as follows: The data used in this study and statistical analysis methods employed are

introduced in Sections 2 and 3, respectively. Our main results are described in Section 4, followed by a discussion in Section 5. The paper concludes with a summary of our main findings.

## 2   Data

The analyses of this study focus on the region from 33°N to 73°N and 25°W to 55°E, encompassing the entire European continent plus some of the surrounding regions of Northern Africa and the Levant. For our calculations, the following data

sources are used:

- LSTAD and NDVI satellite images from the moderate-resolution imaging spectro-radiometer (MODIS) between 2000 and 2015 are retrieved from the NASA Near Earth Observations Program archive (available at http://neo.sci.gsfc.nasa. gov) at a spatial resolution of 0.1°. For both variables, the highest temporal resolution available is chosen, i.e. 16-day intervals for NDVI (NASA, 2016b) and 8-day intervals for LSTAD (NASA, 2016a).

- Land cover data is obtained from the NASA NEO archive at the same spatial resolution of 0.1°. Here, the data of the most recent available year (2011) is used.





– For topographic information, the ETOPO1 Global Relief dataset (Amante and Eakins, 2009) is downloaded at a 1 arc-minute resolution and resampled to a resolution of $0.1°$ by means of spatial averaging.

## 3   Methods

Since we aim to study statistical relationships between the occurrences of extreme events in LSTAD and NDVI (see Section 2)
for different phenophases (Section 3.1), we first identify these events for each grid cell (in the following referred to as a *pixel*) individually (Section 3.2). We then apply *event coincidence analysis* (ECA, Section 3.3) to the resulting pairs of event sequences for each pixel. The resulting *event coincidence rates* are then presented on maps of the study area and additionally evaluated regarding their dependence on land cover type and elevation (Section 3.4).

### 3.1   Phenophases

We subdivide our analysis into different parts of the growing season. This distinction reflects the hypothesis, that a plant's vulnerability to temperature extremes strongly depends on its respective stage of development, i.e. its phenophase (Hatfield and Prueger, 2015). In this study, we consider a rather coarse classification of the growing season into the prevernal, vernal and serotinal phase. Although it would be beneficial to define the relevant season for each pixel within our study area separately (reflecting different climatic conditions and predominant plant types), an investigation of this kind would also require the
consideration of a large number of additional effects influencing local vegetation (e.g. continentality, micro-climate, inter-annual phenophase shifts, etc.). Since such an analysis would extend beyond the limits of the present study, we take here an average continental-scale perspective to identify regional instead of local impact patterns. Therefore, in the remainder of this work, we base our definition of the phenophases on Central European average phenology.

The prevernal phase (early spring) represents the first development of shoots and usually covers the time from March to
April in Central Europe. In the subsequent vernal phase (spring) lasting from May to mid-June, leaves are fully proliferated. The time between July and September can be divided into the aestival (mid-June to mid-July) and serotinal phase (mid-July to September), the latter of which is characterized by the aging of the foliage (Tansley, 1993). As most plants are already fully developed at the start of the aestival phase, we extend here our working definition of the serotinal phase to cover all of July and expand the vernal phase up to the end of June for the sake of simplicity. This results in three different parts of
the growing season investigated in this study: March–April (prevernal phase), May–June (vernal phase) and July–September (serotinal phase).

### 3.2   Event Definitions

#### 3.2.1   NDVI

The temporal resolution of the available NDVI data already determines a minimum time interval of 16 days that an extreme
event would represent. While the investigation of shorter intervals may appear desirable, one has to note that the reaction



time of the NDVI to temperature changes can strongly depend on the type of ecosystem and its resilience or vulnerability in the presence of perturbations (Gonzalez et al., 2010). In this context, NDVI responses to climate extremes can either be instantaneous or show time lags of even up to one month (Tan et al., 2015). Bearing this fact in mind, in addition to the practical necessity of this strategy, defining extreme NDVI events upon a temporal resolution of 16 days appears a reasonable
trade-off to account for both instantaneous and delayed responses at least to a certain degree.

To distinguish between the presence or absence of an extreme event at the given temporal aggregation level, we chose the 10th and 90th percentiles of the available data for each pixel and each phenophase as upper and lower thresholds for defining extremely low and extremely high NDVI values. This approach already accounts for the diverse nature of the very large study area because, e.g., an "extremely low NDVI value" is always defined in relation to the typical distribution of *local* NDVI values
at a given pixel during the phenophase being analyzed. For example, a "low NDVI event"' in a semiarid savanna region might have a very different practical meaning than a "low NDVI event" in a humid temperate forest. Yet, both events are unusual with respect to the local conditions and the current phenophase. By defining the events separately for each phenophase, the seasonal development cycle is taken into account automatically, and no further preprocessing like (practically rather challenging) de-seasonalization of the data prior to further analysis is necessary.

We note that for regions with a dense vegetation cover and a correspondingly high leaf area index during the vegetation period, NDVI is known to follow a saturation curve and to be insensitive to changes in leaf area or biomass at high levels (see studies by, e.g., Huete et al., 1997; Aparicio et al., 2000)). However, this issue mostly applies for high biomass situations like those present in tropical rain forests and is unlikely to affect the results for our study region to a critical extent.

### 3.2.2   Temperature

The definition of the extreme temperature events is performed in the same way as for the NDVI, with the only difference, that the 8-day LSTAD information is averaged to 16-day data first. The thus obtained temperature values do not exhibit any temporal mismatch with respect to the NDVI data. Moreover, as for the NDVI, the pixel and phenophase-wise approach accounts for the spatial and seasonal variability.

### 3.2.3   Event Combinations

The discrimination of the 16-day LSTAD and NDVI data into three phenophases results in time series of 64 (March-April and May-June) or 96 (July-September) data points for each pixel and phase. The 10% and 90% threshold definitions identify 6 (9) low (negative) and 6 (9) high (positive) extreme events per time series. Taken these different types of events in both time series (NDVI and LSTAD) leaves us with four possible event combinations to be considered for each part of the growing season:

1. both LSTAD and NDVI are greater than their respective empirical 90% quantiles (in the following referred to as T90-
V90),

2. both LSTAD and NDVI are lower than their 10% quantile (T10-V10),

3. LSTAD lower than its 10% and NDVI greater than its 90% quantile (T10-V90), and





4. LSTAD greater than its 90% and NDVI lower than its 10% quantile (T90-V10).

In summary, we emphasize that in the present study, the term *extreme* describes values in the tails of the distributions of both considered types of data set rather than record-like events. The limited time span of available satellite measurements results in these extremes also including potentially still relatively moderate seasonal anomalies. However, in the classical peaks-over-threshold sense, it appears reasonable to consider the identified events as extremes.

### 3.3 Event Coincidence Analysis

In order to test for the non-random nature of potential co-occurrences between events in two time series, we apply ECA using the R package `CoinCalc` (Siegmund et al., 2017). ECA computes the empirical fraction of simultaneous events in two series (so-called *coincidences*), which is referred to as the *event coincidence rate* (ECR). By definition, this ECR takes values between 0 and 1, where 0 indicates that the events in both series never occur simultaneously (indicating the absence of a corresponding statistical relationship), whereas an ECR of 1 implies that the events in both series always occur simultaneously. We emphasize that similar approaches have also been used recently by other authors in the context of remote sensing based analyses of ecosystem responses to climatic drivers (Zscheischler et al., 2015).

We assess the statistical significance of the thus obtained ECRs using a simple analytical significance test against the null hypothesis of two independent Poisson processes with low event rates (Donges et al., 2016), using a significance level of $\alpha = 0.05$. Thereby, we obtain spatial and temporal patterns of statistically significant event coincidence rates (SCRs). Specifically, we apply ECA for each joint pixel of the NDVI and LSTAD data individually, resulting in one ECR for each pixel and phenophase.

Note that in this work, we do not further account for possible lagged vegetation responses to temperature extremes. In the latter case, one would have to consider the ECA as a directional analysis tool that distinguishes between the so-called *precursor rate* and *trigger rate* (Donges et al., 2016). In our setting, however, both rates are always the same by definition.

### 3.4 Land Cover and Topography Analyses

In addition to the spatially explicit (pixel-wise) analysis described above, we are also interested in the role of elevation and land cover type as possible covariates determining the vegetation response to temperature extremes.

For the possible effects of elevation, the altitude values of all pixels were grouped into 6 classes defined based on the altitudinal zonation of Frey and Lösch (2014) as follows: planar (below 100 m a.s.l.), colline (100-500 m a.s.l.), submontane (500-1000 m a.s.l.), montane (1000-1600 m a.s.l.), subalpine (1600-2000 m a.s.l.), alpine (2000-3000 m a.s.l.) and nival (above 3000 m). Since only very few pixels fell under the definition of the nival zone, these were combined with the alpine class.

The land cover classification followed the International Geosphere-Biosphere Programme (IGBP) land cover classification scheme (Strahler et al., 1999), excluding the class "water bodies". The shares of the individual land cover classes are summarized in descending order in Tab. 1. As evergreen broadleaf forests and deciduous needleleaf forests make up a negligibly small fraction of the total study area, these two classes have been excluded from the land cover analysis, effectively leaving



14 classes. Furthermore, due to a discrepancy of land-water masks, the land cover image identified some pixels as water area which the NDVI and LSTAD images does not. To account for this data issue, the affected pixels were classified as "no data" and also excluded from the land cover analysis.

## 4   Results

### 4.1   Prevernal Phase (March–April)

Figure 1A,B summarizes the results of ECA for the prevernal phase (March–April). The combination of extremely high temperatures with high NDVI values (T90-V90, green pixels in Fig. 1A) results in SCRs on 14.65% of the terrestrial part of the study area. In Central and Southern Europe, we find spatially contiguous regions of SCRs over mountain regions such as the Alps, the Caucasus and the Carpathians. Further important patches of SCRs exist in large parts of the lowlands of Northeastern Europe (e.g., the Eastern European Plain and Finnish Lakeland) as well as along the southeastern coast of Norway and the coastal regions of Iceland.

In contrast, SCRs between low temperature extremes and extremely low NDVI values (T10-V10, red pixels in Fig. 1B) are widely spread on disconnected patches across large parts of continental Central Europe, the British Isles and Southern Scandinavia. The aforementioned mountain ranges do not appear in this analysis. In total, 13.44% of the study area shows SCRs between low temperature and low NDVI events.

The SCRs between low temperature extremes and extremely high NDVI values (T10-V90, green pixels in Fig. 1B) are sparsely sprinkled across Europe's South, North Africa and Northern Scandinavia, covering only about 2.5% of the total study area. Given the considered confidence level of our significance test, we would accept a false positive rate of 5% in our detected pixel-wise SCRs, implying that the observed pixels with SCRs barely carry any practically relevant information. A similar conclusion applies to the combination between high temperature extremes and extremely low NDVI events (T90-V10, red pixels in Fig. 1A), which also reveals only few pixels with SCRs (3.34% of the study area), mainly within the Fertile Crescent and across the Mediterranean coast of North Africa.

### 4.2   Vernal Phase (May–June)

Unlike the results for early spring, Fig. 1C only shows a small fraction of SCRs for T90-V90 during May and June (green pixels). As an exception, only Scandinavia exhibits a relatively high density of pixels with SCRs, which contribute most to the 5.52% of pixels with SCRs within the total study area.

The combination T10-V10 (Fig. 1D, red pixels) also shows only a relatively low density of pixels with SCRs with the exception of Northwestern Russia along the coastline of the Arctic Ocean (7.80% of the total study area).

In turn, the analysis of T10-V90 during May and June (Fig. 1D, green pixels) reveals SCRs in 9.37 % of the study area. Here, the most important agglomerations can be found in Eastern Europe around the Black Sea, Caspian Sea and Anatolia (especially in its western part).



The most prominent signature of spatially contiguous areas with SCRs during the vernal phase can be observed in Fig. 1C for the analysis of T90-V10 (red pixels). Here, the SCRs cover 16.58 % of the total area and concentrate in five distinct regions: (i) in a belt-like region around the West, North and East of the Black Sea; (ii) along the Fertile Crescent including more northern regions around Azerbaijan; (iii) in a large patch over Southern France north of the Pyrenees; (iv) in more or less the

entire Iberian peninsula (excluding the Pyrenees and the Atlantic coast); and (v) along the Mediterranean coast of the Maghreb regions. We particularly note that over large parts of Wallachia (the flatlands between the Carpathian and the Balkans), the analysis for T90-V10 reveals ECRs of 1.

### 4.3   Serotinal Phase (July–September)

The third considered phenophase only exhibits a negligible number of SCRs over Europe for T90-V90 (Fig. 1E, green pixels).

Only a small agglomeration of SCRs can be found in Northwest Mesopotamia, accounting for just 1.79% of the total study area.

For T10-V10, a broad geographic distribution of SCRs is present (Fig. 1F, red pixels). Notably, these results resemble the observations for the same combination in May–June to a certain degree: the SCRs are loosely spread in the Scandes and Northwest Russia (6.88% of the total study area).

The co-occurrence patterns of T10-V90 (Fig. 1F, green pixels) are concentrated in a latitudinal belt (approximately between 40 and 45°N) from the Caspian Depression via the Kuma-Manych Depression (North of the Caucasians), the Balkans, almost entire Italy to parts of the Iberian peninsula. The Caucasians and the Alps appear to be distinctively excluded from this pattern. In total, 21.93% of the study area shows SCRs for T10-V90.

Finally, the combination T90-V10 (Fig. 1E, red pixels) exhibits widespread SCR pattern, which cover almost all of Western,

Central and Eastern Europe. Especially high ECRs are found between the Adriatic and the Black Sea, while remarkable areas without SCRs include major mountain ranges like the Alps, the Pyrenees, the northern Carpathians and the Caucasus Mountains. For this combination, 35.52% of the total study area is covered by SCRs.

### 4.4   Topographical Effects

Figure 2 illustrates the distribution of pixels exhibiting SCRs (for every combination of events and all three phases) among the

six elevation classes. The width of the displayed bars indicates the fraction of pixels with SCRs among the total study area and therefore (to a certain degree) underlines the relevance of this event combination during the respective phenophase. The SCRs for the two event combinations T90-V90 and T10-V10 during the prevernal phase (Fig. 2, left panel) mostly occur on elevation levels of the planar and colline zone, while T10-V10 has a slightly higher tendency towards higher elevations than T90-V90. The alpine zone also exhibits a notable number of pixels with SCRs as compared to the other two event combinations of the

prevernal phase. When comparing this to Fig. 1A,B, the pixels of the alpine zone seem to mainly result from event coincidences in the European Alps for T90-V90 and the Pyrenees and Caucasian Mountains for T10-V10. The few significant pixels of the other two event combinations (T90-V10 and T10-V90) are evenly distributed throughout all elevation classes but the alpine zone.



For the vernal phase (Fig. 2, central panel), the largest differences between the event combinations can be found in the planar zone, where T10-V90 exhibits SCRs at a more than twice as large fraction of pixels as compared to T90-V90. Notably, the event combination T10-V90 shows a clearly higher contribution at the subalpine and alpine zone than the other event combinations, which presumably mainly results from the SCRs in western Anatolia (see Fig. 1D).

The two event combinations showing the most SCRs during the serotinal phase (T90-V10 and T10-V90, right panel of Fig. 2) again contain the most pixels in the lowest two elevation zones. Yet, T10-V90 has a distinctively larger fraction of pixels with SCRs in the higher four zones than T90-V10, especially in the alpine zone.

### 4.5 Land Cover Effects

The results for the analysis of the underlying land cover classes of the SCRs are summarized in Fig. 3. Like for the topographical
analyses, the fraction of pixels exhibiting SCRs is plotted in stacked bars, where the width of the bars reflects the share of significant pixels on the entire study area.

During the prevernal phase (Fig. 3, left panel) SCRs of the event combinations T90-V90 and T10-V10 mainly occur on pixels classified as mixed forests and evergreen needleleaf forests as well as the two cropland classes. The fraction of pixels with SCRs on forests is clearly larger for T90-V90 than for T10-V10, which is replaced in T10-V10 by grassland and cropland.
The two other event combinations, which exhibit much fewer SCRs in general, are dominated by open shrubland and cropland.

During the vernal phase (Fig. 3, central panel) the SCRs of the event combinations T90-V90 and T10-V10 are relatively evenly distributed among mixed and evergreen needleleaf forests, woody savanna, open shrubland and grassland. In contrast, the SCRs of T90-V10 and T10-V90 almost only occur on pixels classified as cropland or grassland, where for T10-V90 grassland plays a much more important role than for T90-V10.

Finally, for the serotinal phase (Fig. 3, right panel) the dominant event combinations T90-V10 and T10-V90 show a picture similar as for the vernal phase, yet with a distinctively higher fraction of SCRs on mixed forests.

## 5 Discussion

### 5.1 Prevernal Phase

The most important finding for the prevernal phase is that the strongest impacts of temperature extremes appeared for the
two event combinations V90-T90 and V10-T10 (joint positive and negative extreme values, respectively). This suggests a generally positive statistical relationship between LSTAD and NDVI in early spring. Although temperature is widely known to be among the most crucial factors for most plants' physiological regulation during this early period of the year (see, e.g., Summerfield and Roberts, 1988; Srikanth and Schmid, 2011), it is surprising that the SCR signatures in Fig. 1A,B only cover comparatively small parts of the study area. In more detail, most SCRs for V90-T90 occur in regions which normally exhibit
low temperatures (as compared to the rest of the study area) at that time of the year: Northern Europe and high mountain ranges such as the Alps and the Caucasus Mountains. The topographical analysis (Fig. 2) confirms the increased relevance of



higher elevation classes. Similar observations have been made by Cannone et al. (2007), who found fast growth responses of alpine vegetation to increasing temperatures, since the length of the snow cover season decreased. At the same time, earlier bud burst at these altitudes increases the vulnerability of the vegetation to freezing events after a short window of warm conditions (Wheeler et al., 2014).

The land cover analysis suggests that especially croplands, mixed forests, evergreen needleleaf forests and grasslands benefit from extraordinary warm conditions during early spring. On the other hand, also negative effects of cold periods are mainly visible on croplands in the Eastern European Plain (chiefly wheat and corn fields) and Central Europe (fruit trees, field crops). Being adapted to a narrow range of climatic conditions, most of these croplands have been reported to be vulnerable to late spring frost events and might be at even larger risk when the growing season extends earlier into the year (Chmielewski et al.,
2004; Lavalle et al., 2009; Trnka et al., 2014).

Although the patterns for the other two event combinations are not as pronounced as for the aforementioned two, three distinct regions showed very specific behaviors: the Scandes, the Atlas Mountains and Mesopotamia (see Fig. 1A,B). The dominant land cover class in these three regions is open shrubland (see Fig. 3), which the IGBP classifies as *"lands with woody vegetation less than 2 meters tall and with shrub canopy cover between 10-60%. The shrub foliage can be either evergreen*
*or deciduous"* (Strahler et al., 1999, p. 17). This sparse canopy cover leaves the soil prone to a loss of moisture during high temperature events, which may in the course damage non-perennial grass seedlings that heavily depend on moist conditions in the top soil layer (Harrington, 1991). While this phenomenon can already be observed in spring in these regions, it appears throughout the season and seems to reach its peak in the vernal phase.

Already at this point, it is very important to note, that the SCR patterns of T90-V90 and T10-V10 (as well as those of T10-
V90 and T90-V10) do not match in many cases. This implies, that in many parts of Europe, a positive or negative relationship between temperature and NDVI is only valid for one tail of the empirical distribution, which presents a finding that could not be deduced based upon classical linear correlation analyses (e.g. Schultz and Halpert, 1995; Los et al., 2001; Ichii et al., 2002; Wang et al., 2003; Stöckli and Vidale, 2004; Hao et al., 2012).

## 5.2  Vernal Phase

For the vernal phase, SCRs for the event combination T90-V90 exhibit a distinct pattern in the northern Scandes. When comparing these results to the same combination during the preveral phase, the observed phenomenon appears to have shifted northwards, presumably due to the later onset of spring in these latitudes. On the other hand, SCRs of T10-V10 are particularly frequent in the northeastern parts of the study area. As mentioned before, this observation may be due to damages induced by cold spells during bud burst or, more likely, late snowfall events that hide the sparse tundric vegetation cover (Ellenberg and
Leuschner, 1996) from being seen by MODIS.

The largest amount of SCRs appeared for T90-V10. While this heat stress situation is found relevant for vast parts of Europe, croplands and partly grasslands were the most affected land cover types. One reason for this particular sensitivity could be the following. As crops are meant to produce a maximum yield, they often demand high standards in nurturing and water availability and are vulnerable to sudden changes (Semenov and Shewry, 2011; Ma et al., 2015). As already discussed





in the introduction, extreme temperatures can cause substantial losses in crop yields (see, e.g., Schaap et al., 2011; Lesk et al., 2011; Moriondo et al., 2010; Wreford and Adger, 2010). Still, regional disparities appeared. For example, the Wallachia proved to be particularly prone to heat stress. Yet, at the same phenophase other lowlands at similar latitudes like the Padan Plain in Northern Italy did not exhibit such a behavior. For most parts, we suggest a different water supply situation of these areas (e.g.,

influenced by continentality, size of the watershed and irrigation) to lead to this finding, while also differing water demands of the vegetation due to different dominating crop types may play a role.

Regarding the repeated appearance of grasslands among the most affected land cover classes for the combination T90-V10, we note that a rapid loss of soil moisture during high temperature events is expected to result in negative NDVI events (Teuling et al., 2010).

## 5.3   Serotinal Phase

The combination T90-V10 resulted in the highest fraction of SCRs for the serotinal phase. This result suggests that extremely high temperatures pose a severe threat to the vegetation particularly during this time of the year. Notably, three of the most severe heat waves on record (in the summers of 2003, 2006 and 2010) occurred during our study period (2000-2015) and most likely constitute a main part of the selected upper 10% of temperature values. Previous studies from these summers

confirm largely adverse effects on vegetation throughout Europe, with regionally higher intensities along the Mediterranean coast (De Bono et al., 2004; Ciais et al., 2005; Fischer and Schär, 2010).

While heat waves themselves already pose a considerable challenge to plants, the meteorological conditions preceding these events also need to be considered as they may exacerbate the impacts of heat on vegetation. The 2003 heat wave provides a good example, as it was preceded by unusually dry conditions between February and August (Fischer et al., 2007), which left

the vegetation at severe drought stress even before the actual extreme event. Hence, the high share of pixels with SCRs may also result from vegetation, which had already been stressed in previous months and was thus more vulnerable to repeated temperature extremes.

Interestingly, like in May–June, croplands and grasslands were among the most affected land cover types. These results match well with modeled future crop production (Deryng et al., 2014; Teixeira et al., 2013), which also sees Southeastern and

Eastern Europe at a particularly high risk of crop failure due to extreme temperature events in late spring and summer. Still, it needs to be noted, that during this time of the year parts of the extremely low NDVI events may also be the result of harvesting activities (Wardlow et al., 2007). For example, in Ukraine winter wheat harvest usually takes place between the end of June and mid-August (USDA, 2016), which would be compatible with the low NDVI values present in this region during the vernal and serotinal phase.

At the same time, SCRs of T10-V90 were found in large areas around the Black and Mediterranean Seas. This could be interpreted as a "pause for breath" for crop species that are under continuous stress during that period of the year. In contrast, the high fraction of grassland pixels among all SCRs of this event combination contradicts this idea, since the composition of grass species can be assumed not to be anthropogenically influenced and thus, to be well adapted to the local climate. Another explanation could be, that the cold temperature events during this phenophase also coincide with periods of high precipitation,



and that the latter (or a combination of both factors) is actually causing the increase in NDVI. This hypothesis shall be further tested in future work.

On a final note, it shall be pointed out, that the analysis of altitudinal zones yielded no clearly interpretable result. The distribution of SCRs among the different classes more or less represented their parent distributions over the whole study area. This

outcome may largely be owed to the fact, that the altidutinal classes were defined uniformly across Europe. That said, especially when looking at the largest European mountain ranges in isolation, the Alps, Carpathians, Pyrenees and Caucasus mountains indeed showed unique response patterns, which highlights their exceptional position among the European landscapes.

## 6    Conclusions

In the context of projected increasing frequencies of temperature extremes in Europe (Coumou and Rahmstorf, 2012; Tank

and Konnen, 2003; Luterbacher et al., 2004; IPCC, 2013; Barriopedro et al., 2011; Petoukhov et al., 2013; Seneviratne et al., 2012), the present study has delivered a spatially resolved statistical assessment of the impacts of very high and very low temperature events on European vegetation. By applying event coincidence analysis to high-resolution remote sensing data, we have investigated the co-occurrences of extremes in daytime land surface temperature anomalies and the normalized difference vegetation index over a period of 16 years (2000-2015). We have analyzed patterns of significant local event coincidence

rates, accounting for the type of land cover and altitudinal zone and assessing which regions might be especially vulnerable to possibly rising frequencies and/or intensities of future extreme temperature events.

Our results revealed, that the vulnerability to heat stress (particularly between May and June) is very heterogeneously distributed over Europe. The time between July and September displayed the highest densities of significant event coincidence rates for both, high and low temperature extremes. This was especially true for Southern Europe, which emphasizes the vul-

nerability of this region to climate extremes also in the context of expected increasing frequencies of future heat waves.

Our analysis of the distribution of significant event coincidence rates among altitudinal classes did not reveal clear results. From a geographical point of view, however, high mountain ranges like the Alps and the Caucasus Mountains formed a visible pattern, which points to unique responses of alpine vegetation as compared to the main patterns across Europe. In contrast, the analysis for the different land cover types revealed, that the areas suffering from both low and high temperature

extremes are mostly anthropogenically shaped landscapes. These ecosystems appear to be particularly sensitive to temperature extremes. Another important finding is that forests hardly exhibited significant event coincidences rates, which indicates that these vegetation types are more resilient against temperature extremes than others. An important exception are mixed forests, which clearly benefit from warm temperature events during spring.

Despite the reported achievements, further research on simultaneous occurrences of climatic extremes with vegetation-

related extremes is strongly recommended. On the one hand, an explicit consideration of moisture-related variables (like precipitation, soil moisture or drought indices) within the framework of event coincidence analysis would surely yield further valuable insights (for example, for differentiating between heat and drought stress). On the other hand, an investigation of the vegetation vitality and atmospheric conditions preceding extreme temperature events could clarify the actual impact of a



specific event itself on vegetation. To this end, we emphasize the possibility of employing previously developed multivariate extensions of the present analysis methodology (Siegmund et al., 2016b) for the aforementioned purposes. We outline such investigations as a subject of future work.

**Acknowledgments.**

5  This work has been financially supported by the German Federal Ministry for Education and Research (BMBF) within the framework of the BMBF Young Investigators Group CoSy-CC$^2$: Complex Systems Approaches to Understanding Causes and Consequences of Past, Present and Future Climate Change (grant no. 01LN1306A). JFS acknowledges funding by the Evangelisches Studienwerk Villigst e.V. The authors express their gratitude to Catrin Gellhorn and Chiranjit Mitra for giving valuable feedback to this work.

10  **Code availability**

All calculations in this work have been based upon the open source R package `CoinCalc`.

**Data availability**

All data used in this work are publicly available at the sources described in the main text.

*Author contributions.* JFS designed the analysis. LB, JFS and MM conducted the analysis. LB and JFS prepared the manuscript. RVD
15  supervised the analysis and revised the manuscript and the interpretation of the obtained results.

*Competing interests.* The authors declare no conflict of interest.



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



**Table 1.** Share of each land cover class on the total study area. Pixels classified as "no data" cover 1.23% of the area.

| IGBP Land Cover Units | areal share [%] |
| --- | --- |
| Croplands | 25.190 |
| Mixed Forest | 20.870 |
| Grasslands | 14.245 |
| Cropland/Natural Vegetation Mosaic | 9.318 |
| Woody Savannas | 8.279 |
| Open Shrublands | 7.022 |
| Evergreen Needleleaf Forest | 5.913 |
| Barren or Sparsely Vegetated | 4.118 |
| Deciduous Broadleaf Forest | 1.362 |
| Permanent Wetlands | 0.826 |
| Urban and Built-Up | 0.819 |
| Permanent Snow and Ice | 0.504 |
| Savannas | 0.274 |
| Closed Shrublands | 0.017 |
| Evergreen Broadleaf Forest | 0.006 |
| Deciduous Needleleaf Forest | 0.002 |





**Figure 1.** Spatial patterns of significant event coincidence rates (SCRs) between different combinations of low (left panels) and high (right panels) temperature and NDVI extremes during the three considered phenophases, derived from MODIS satellite measurements from 2000 to 2015. Red colors show SCRs between temperature-related extremes and extremely low NDVI values, green colors such for extremely large NDVI values. Grey areas indicate pixels with non-significant event coincidence rates.




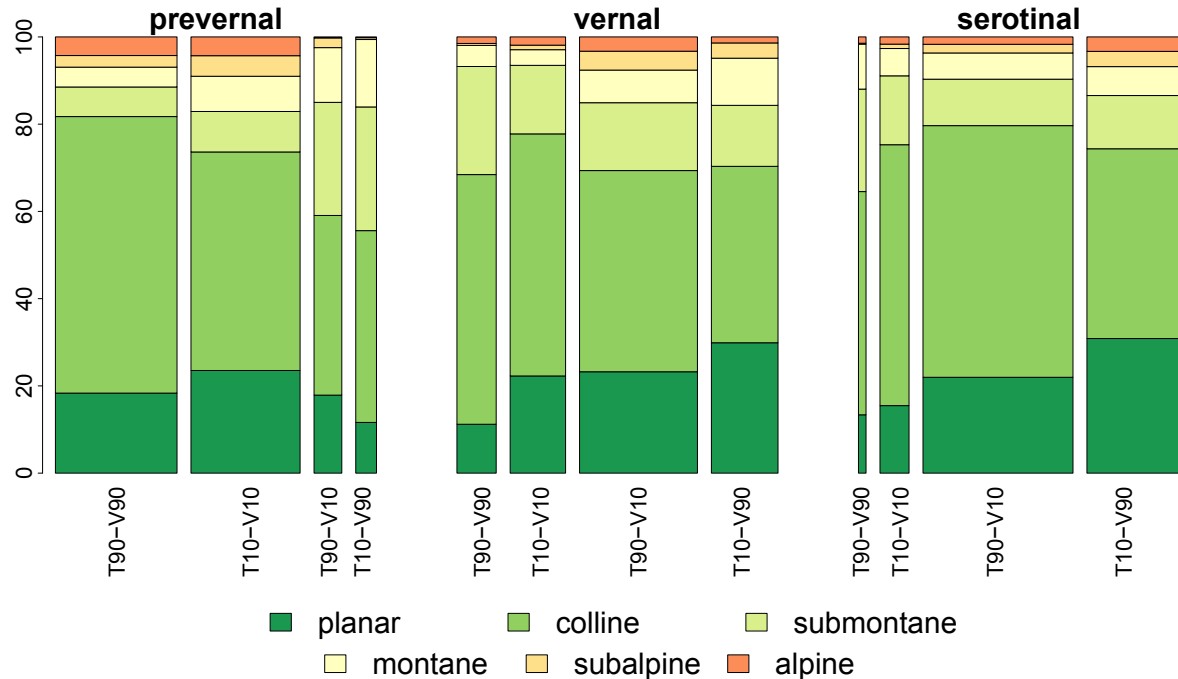

**Figure 2.** Distribution of pixels with SCRs among the seven altitudinal classes (in percent) for all four event combinations in the three phenophases. The width of the columns represents the fraction of pixels with SCRs of the specific event combination within the total study area.





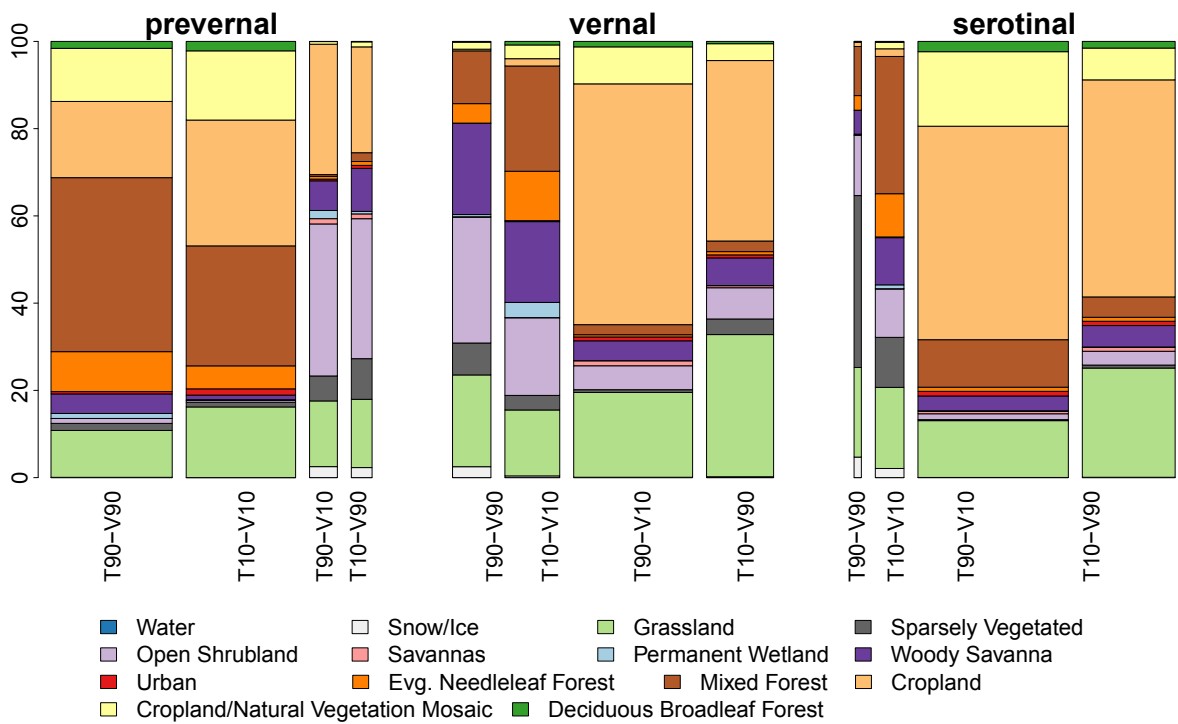

**Figure 3.** Same as in Fig. 2 for the 14 land cover classes.