# Peer review of "Impacts of temperature extremes on European vegetation during the growing season"

_Biogeosciences, 2017_

## Referee Comment (RC1) · G. Moser (Referee) · 6 Jul 2017

G. Moser (Referee)

gerald.moser@bio.uni-giessen.de

General Comments: A very nice contribution to BG that shows significant effects of weather extremes on vegetation development in Europe. The application of event co-incidence analysis to high-resolution remote sensing data points out the impacts of temperature extremes on NDVI and revealed that the vulnerability to summer heat stress is not homogeneously spread, but concentrated in different regions of Europe and particularly high for specific vegetation types.

Specific comments: There are only few points that could/should be improved: Please, don't call your distinguished periods of the data sets after phenophases, because the continental averages are nearly everywhere wrong except for central Europe. Under

3.3 in the last three sentences you describe the possibility to include the analysis of lagged vegetation responses to extreme events. But you only state, that you did not consider possible lag times – I wonder why? I think, this would have been very interesting. So please, give a proper reason why you didn't account for it, or reanalyse the data once more considering potential lag times.

Only in the discussion of the last period ("serotinal phase") you mention the years when extrem events occurred. I would really appreciate if you could include these informations for all periods and all types of SCRs in the discussion or even better in the results section.

I wonder why you cite Ellenberg 1996 (Ellenberg & Leuschner is only correct for the lates edition in 2010) text book about central Europe, as you referred to Northern Europe, where Dierßen 1996, Vegetation Nordeuropas matches much better.

I was surprised, when you mentioned the problem that harvesting activities might create biases in your results. I expected that for the mentioned harvesting activities in Ukraine, the example given on page 5, lines 6-14, for the semiarid savanna region would mean that harvested areas have NDVI close to the local mean value and would not be indicated as a low NDVI pixel. So, maybe I misunderstood that, but I hope you can explain that.

Technical comments: Clarify the meaning of the numbers in asterisks in line 26 and 27 on page 5

---

## Referee Comment (RC2) · Anonymous Referee #2 · 18 Jul 2017

In this manuscript, the authors employ event coincidence analysis (ECA) to quantify the likelihood of simultaneous occurrences of extremes in daytime land surface temperature anomalies and the normalized difference vegetation index (NDVI) over entire Europe, and reveal the spatio-temporal patterns of occurrence between temperature and NDVI. Generally, this is an excellent manuscript. The background of the problem has been well presented, the detailed aspects of the data and approach have been introduced and discussed, the results and discussions are adequate and well organized, the conclusions are to the point. So I recommend to publish this manuscript.

The following comments are just for reference.

1. It is interesting to see the one tail feature, which cannot be deduced from traditional linear correlation analyses, and a little out of the intuitive. But how this feature can be

validated? To device more validation work, direct or indirect, field measurements or social impact, may help in the future work.

2. There are quite some versions of NDVI and temperature datasets, as 7 NDVI products mentioned in Scheftic et al., 2014, how other dataset would perform? Scheftic, W., Zeng, X., Broxton, P., and Brunke, M. 2014. Intercomparison of seven NDVI products over the United States and Mexico. Remote Sens. 6:1057-1084, doi:10.3390/rs6021057.

3. To address the event, the authors used the percentile applied to NDVI and temperature, which is quite similar to the vegetation health approach adopted in Kogan, 2001. Kogan, FN. 2001. Operational space technology for global vegetation assessment. Bull Am Meteorol Soc. 82:1949–1964.

4. typo, Page 16, Line 13, same url was written twice.

---

## Referee Comment (RC3) · Anonymous Referee #3 · 1 Aug 2017

The paper addresses the important issue of plant responses to very hot or very cold periods. As the investigated region is rather large, the use of remote sensing data is the only feasible option, and the choices made here are among the most standard ones. Still, alternative to NDVI are debated, e.g. NPP (which would be, of course, model results rather than directly based on optical imagery) might reflect the vegetation productiviity better. The authors do not discuss why they are using daylight temperature anomalies (LSTAD) rather than simply average air temperatures. It is not sure whether this would make much of a difference, but did the authors even try? Temporal resolution is an important issue as well. While the advantage of using 16-days aggregatets is probably that lag effects are of minor importance (and, thus, not considered at all by the authors), an obvious disadvantage is that the concept of an "event" is blurred

- with respect to meteorological phenomena, this time scale is so much longer than the synoptic scale that virtually no meteorologist would consider the averaged values as "events". The spatial resolution is high, but probably still too course to conclude reliably on topographical or land cover covariate effects - and the authors do not find any clear results for them, accordingly. The concept of an "extreme" event usually refers (in POT approaches) to quantiles (much) higher than P90 or lower than P10. The wording should be rephrased to "high temperatures", "low NDVI" and so on, not even "very high" etc. An obvious extension of the approach would be to investigate the impact of changing the percentile thresholds on the spatial patterns obtained - are the conclusions basically the same when trying P80/P20 or P95/P05 ? You cannot go much further than P95 since the number of events present in the time series gets too low. It is also a weak point that by construction every pixel contains the very same amount of "extreme" events. That the heatwave of 2003 did affect only some parts of Europe vanishes from the analysis thereby, and, as the authors rightly point out, for rather stable regions w.r.t. temperature fluctuations and NDVI (think of evergreen forests for example) there is nothing particular or unusual with the values exceeding P90. The distinction of the phenophases is a clear plus for the analysis and, although not unexpected, reveals that the coincidence of the four different combinations of low and high values for LSTAD and NDVI differs very much between the phases. Still, to use the same calendar dates to differentiate the phases is very simplistic and could be improved a lot, given the accumulated knowledge about regional budburst and senescence timings for the different part of Europe. Still, the results quantify nicely the intuitive expectation and demonstrate clearly the non-linearity of the relationship. ECA is in this case without doubt superior to conventional correlation analysis. Another obvious and probably very necessary extension is the consideration of moisture effects, e.g. simply using precipitation time series. This is discussed towards the end of the paper. The presentation is rather clear and of appropriate length. Some of the objections the reviewer has are also considered by the authors, making their contribution rather balanced, no overstating of their results. The suggestions made in this review accumulate

to not more than a minor revision.

---

## Author Comment (AC1) · 12 Sep 2017

We are very grateful for the positive overall evaluations and helpful comments raised by the reviewer, which we would like to take as a basis for a thorough revision of our manuscript and encouragement for further work. A detailed point-by-point response, together with answers to the questions raised by the other referees, is provided as a supplementary file accompanying this author comment.

Please also note the supplement to this comment:
https://www.biogeosciences-discuss.net/bg-2017-189/bg-2017-189-AC1-supplement.pdf

[Figure]

[Figure]

**Supplement:**

Dear Editor,

We are grateful to all three reviewers of our paper for their careful evaluation of our manuscript and the numerous helpful recommendations for improving the presentation of our results. On completion of the interactive discussion phase, in the following we present our answers to the referee comments. Green text colour highlights suggested changes and additions to the manuscript text.

Best regards,

Lukas Baumbach, Jonatan F. Siegmund, Magdalena Mittermeier and Reik V. Donner
* * *
**Response to Referee 1 (Gerald Moser, 06 July 2017):**

**Remark 1:**
*"Please don't call your distinguished periods of the data sets after phenophases, because the continental averages are nearly everywhere wrong except for central Europe."*

**Answer:**

We agree that the division of our analysis by phenophases is potentially misleading when considering the heterogenity of seasonal development stages across Europe.  We therefore suggest the term "phenophase" from the vast body of our manuscript and use it only as a motivation to explain the origin of our subdivisions. In all other instances, "phenophase" will be replaced with "time period". Specifically, we will modify the heading of Section 3.1 and add a few sentences along the following lines:

> "Therefore, we base the subdivision of the analysis on Central European average phenology (Ellenberg and Leuschner, 2010).  The chosen time intervals of March-April, May-June and July-September here correspond to the prevernal, vernal and serotinal phases of Central European vegetation. […] We emphasize, that this classification is only accurate for Central Europe, while the phenophases differ in onset and length in other parts of Europe. The subdivision should thus be understood as a starting point for differentiating between impacts during different seasons but does not allow for a systematic inter-comparison between vegetation responses to climate stress among different parts of Europe from the plant physiological perspective."

**Remark 2:**

*"[…] you describe the possibility to include the analysis of lagged vegetation responses to extreme events. But you only state, that you did not consider possible lag times – I wonder why? I think, this would have been very interesting. So please, give a proper reason why you didn't account for it, or reanalyse the data once more considering potential lag times."*

**Answer:**

By our definition of NDVI events, one "event" represents a strong anomaly in the averaged NDVI taken over 16 days. In this spirit, a possible lagged responses can already be included in such a time window with a considerable probability (as also noticed by reviewer 3), since they arise commonly relatively fast after a climate-related disruption (i.e., less than a month after that event). Considering a lag effect in our case is therefore unlikely to add valuable information to the present study, since its

temporal resolution is too coarse. However, we agree that additional analyses on a daily basis would be worth striving for in follow-up studies.

To further clarify this point in our manuscript, we suggest adding a sentence at the end of Sect. 3.3:

> "Note that in this work, we do not further account for possible lagged vegetation responses to temperature extremes due to the coarse temporal resolution of our data, which may result in time windows capturing both instantaneous as well as lagged responses. Methodologically speaking, if an additional time lag were included, one would have to consider the ECA […]"

**Remark 3:**

*"Only in the discussion of the last period ("serotinal phase") you mention the years when extrem events occurred. I would really appreciate if you could include these informations for all periods and all types of SCRs in the discussion or even better in the results section."*

**Answer:**

We agree that further details on the other two phases might be useful, and therefore suggest adding the following additional information to the discussion section:

> "For example, in 2007 the Northern Europe Wheat Belt (stretching from Northern France to the Baltic) experienced a spring backlash after an unusually warm winter followed by a dry and cold spring (USDA, 2007). As a result, heat-advanced crops were damaged throughout Northern Central Europe, which is also visible in the results of Fig. 1B." (Section 5.1 March-April)

> "For instance, in 2011 France experienced one of the warmest springs on record which resulted in a 12% decrease of grain harvest (Coumou and Rahmstorf, 2012; NOAA, 2012). Similarly, based on our results (Fig. 1C), the Wallachia proved to be particularly prone to heat stress. This corroborates observations of Spinoni et al. (2015), who identified almost annually recurring heat waves, with the most extreme situations occurring in 2003 and 2007. However, other lowlands at similar latitudes like the Padan Plain in Northern Italy did not exhibit such a behavior." (Section 5.2 May-June)

Providing further examples (or just more extensive list of cases) beyond these few cases would in our opinion render the manuscript unnecessarily long. Therefore, we suggest refraining from adding more than the material mentioned above.

**Remark 4:**

*"I wonder why you cite Ellenberg 1996 (Ellenberg & Leuschner is only correct for the lates edition in 2010) text book about central Europe, as you referred to Northern Europe, where Dierßen 1996, Vegetation Nordeuropas matches much better."*

**Answer:**

The updated edition of Ellenberg & Leuschner 2010 should indeed replace the version of 1996 as reference for our approach to distinguishing between phases. We would suggest adding Dierßen 1996 as reference to the description of tundric vegetation.

**Remark 5:**

*"I was surprised, when you mentioned the problem that harvesting activities might create biases in your results. I expected that for the mentioned harvesting activities in Ukraine, the example given on page 5, lines 6-14, for the semiarid savanna region would mean that harvested areas have NDVI close to the local mean value and would not be indicated as a low NDVI pixel. So, maybe I misunderstood that, but I hope you can explain that."*

**Answer:**

It is true that the extreme values are defined locally. However, this does not exclude the possibility, that they may be caused by different factors (stress, harvest, land use change, etc.). In case that harvesting activities result in similarly low NDVI values during all considered years of observations, the situation could arise that during this time of the year no stressed vegetation is identified by our analysis at all. Based on the data and method used in the present study, there is no safe way of separating different cases, which is a classical drawback when working with remote sensing data. The actual underlying mechanism might be revealed through validation with additional ground-based data, which is, however, beyond the scope of the present work.

In the context of this situation, we suggest adding the following paragraph to our manuscript in order to clarify how NDVI events may be influenced through confounding factors (Sect. 3.2.1):

> "It should also be noted, that changes in the NDVI may be attributed to different underlying mechanisms, which may not be distinguished safely through analysis of remote-sensing based observations. While in this work, we specifically analyze the co-occurrence of very atypical NDVI values with episodes of extraordinary temperature anomalies, other factors including human activities like harvests or land use changes may also lead to NDVI anomalies, which need to be considered for an appropriate interpretation of the results."

**Remark 6:**

*"Clarify the meaning of the numbers in asterisks in line 26 and 27 on page 5"*

**Answer:**

We assume that the referee is referring to the numbers in brackets in the indicated lines. We suggest rephrasing this sentence as follows:

> "Accordingly, for the time intervals March-April and May-June, the 10% and 90% threshold definitions identify 6 low (negative) and 6 high (positive) events per time series, while for July-September, 9 low and 9 high events are selected."
* * *
**Response to Anonymous Referee 2 (18 July 2017)**

**Remark 1:**

*"It is interesting to see the one tail feature, which cannot be deduced from traditional linear correlation analyses, and a little out of the intuitive. But how this feature can be validated? To device more validation work, direct or indirect, field measurements or social impact, may help in the future work."*

**Answer:**

Unfortunately, it is not fully clear to us what precisely the reviewer refers to here when asking for the validation of a certain "feature". The term "validation" is commonly used with respect to data sets or results of a specific type of analysis, though with different meanings in different communities (see Loew et al. (2017), Sect. 2.1, for a corresponding discussion). We agree that the presented results would be considerably strengthened when being (qualitatively) recovered using other data sets (both ground and satellite based) as well as alternative methodologies. To our knowledge, however, there are no other statistical approaches that could be equivalently used for capturing the property quantified by the framework used in our manuscript based on the same (relatively short) data sets. We will explicitly mention this fact in our revised manuscript.

**Remark 2:**

*"There are quite some versions of NDVI and temperature datasets, as 7 NDVI products mentioned in Scheftic et al., 2014, how other dataset would perform? Scheftic, W., Zeng, X., Broxton, P., and Brunke, M. 2014. Intercomparison of seven NDVI products over the United States and Mexico. Remote Sens. 6:1057-1084, doi:10.3390/rs6021057."*

**Answer:**

We agree that there are numerous other variants of NDVI (and related vegetation indices) as well as temperature data sets, and that it would be worthwhile to study the robustness of our findings based upon an inter-comparison taking various complementary data sets and variables into account. Since different data sets should highlight similar dynamics, we would expect that corresponding investigations should yield qualitatively similar results. However, this has not been systematically tested by us so far. To this end, we have chosen to use only two specific (and well-studied) data sets, leaving the task of repeating the presented analyses with other data sets a subject of future work.

We suggest including a comment on this possible extension of our approach in the conclusion section:

> "Finally, an investigation of different datasets on vegetation stress is recommended to study the robustness of our approach."

**Remark 3:**

*"To address the event, the authors used the percentile applied to NDVI and temperature, which is quite similar to the vegetation health approach adopted in Kogan, 2001. Kogan, FN. 2001. Operational space technology for global vegetation assessment. Bull Am Meteorol Soc. 82:1949–1964."*

**Answer:**

We thank the reviewer for this interesting comment. While Kogan also analysed extreme values of NDVI and temperature over a relatively long time period (14 years), there are clear differences to our approach. Specifically, Kogan calculated min-max conditions over all pixels and integrated several proxies (NDVI, moisture, brightness temperature to a vegetation health index. In turn, our analysis works pixel-wise and solely uses the NDVI. We will add a brief note on this aspect to the conclusion section of our revised manuscript:

"On the one hand, an explicit consideration of moisture-related variables (like precipitation, soil moisture or drought indices) within the framework of event coincidence analysis would surely yield further valuable insights (for example, for differentiating between heat and drought stress, see also (Kogan 2001)."

**Remark 4:**

*"typo, Page 16, Line 13, same url was written twice."*

**Answer:**

We will correct this error in our revised manuscript.
* * *
**Response to Anonymous Referee 3 (1 August 2017)**

**Remark 1:**

*"The authors do not discuss why they are using daylight temperature anomalies (LSTAD) rather than simply average air temperatures. It is not sure whether this would make much of a difference, but did the authors even try?"*

**Answer:**

In our opinion, there are multiple reasons for not using average air temperatures. Most of all, we are investigating (relative) extreme events during the growth period of European vegetation. In this situation, we may expect that during most parts of their growing season, plants react more strongly on heat stress (positive temperature anomalies, mostly expressed during daytime) than on cold anomalies (commonly occurring during night-time). This calls for focusing on daytime temperatures, since this variable captures the daily active time (photoperiod) or most plants. In turn, we intentionally exclude the effects of overnight frost damages especially in spring from our analysis. According to this rationale, using daily-average air temperatures (i.e., averaging temperatures over their full diurnal cycle, including both, day- and night-time values in addition to the averaging over the 16-day periods) may hide possible extreme conditions during the photoperiod. In order to clarify this aspect, we suggest adding the following sentence in the introduction to illustrate our rationale:

"A basic problem related to analyzing such impacts lies in the definition of temperature extremes. One way to identify extraordinary temperatures is to investigate anomalies of long-term average temperatures during specific time intervals. Therefore, land surface temperature anomalies during daytime (LSTAD) represent a suitable observable for the purpose of this study, since they integrate temperature information over the daily active time (photoperiod) of most plants."

**Remark 2:**

*"Temporal resolution is an important issue as well. While the advantage of using 16-days aggregates is probably that lag effects are of minor importance (and, thus, not considered at all by the authors), an obvious disadvantage is that the concept of an "event" is blurred - with respect to meteorological phenomena, this time scale is so much longer than the synoptic scale that virtually no meteorologist would consider the averaged values as "events"."*

**Answer:**

We understand this concern, although there are counter-examples in the meteorological literature (e.g., the term "drought event" sometimes used instead of "drought episode" refers by definition to an extended period of time beyond normal synoptic time scales). In our present work, the term "event" is not to be understood in a meteorological sense, which is why we dedicated a whole subsection solely to the clarification of this term. We suggest adding the following sentence at the beginning of section 3.2.1 for further clarification:

> "We emphasize that the notion of the term "event" as used in the present work should therefore be understood in a statistical, not a synoptic-scale meteorological sense."

**Remark 3:**

*"The spatial resolution is high, but probably still too coarse to conclude reliably on topographical or land cover covariate effects - and the authors do not find any clear results for them, accordingly."*

**Answer:**

While the resolution may not be high enough for clear local interpretations, as mentioned in the introduction of the paper, our main focus lies on identifying regional or larger scale patterns. Since the same resolution was used for all remote sensing data (including land cover and elevation), the detection of covariate effects is still possible at a regional scale. The ambiguity of our results, however, is more likely to originate mostly from other factors like classification criteria for land cover (which is not tailored to Europe), influence of moisture, microclimate etc.

**Remark 4:**

*"The concept of an "extreme" event usually refers (in POT approaches) to quantiles (much) higher than P90 or lower than P10. The wording should be rephrased to "high temperatures", "low NDVI" and so on, not even "very high" etc. An obvious extension of the approach would be to investigate the impact of changing the percentile thresholds on the spatial patterns obtained – are the conclusions basically the same when trying P80/P20 or P95/P05 ? You cannot go much further than P95 since the number of events present in the time series gets too low."*

**Answer:**

Indeed, we tested different quantiles before our final analysis and found that the results did not change qualitatively when using, for example, P15/85 or P05/95. However, as the referee rightly points out, the use of 5% tails results in a very small number of "events", which forms a weak and uncertain basis for our analysis. The word "extreme" may appear misleading at a first glance, however, we describe our rationale for still using it in the last paragraph of section 3.2.3:

> "In summary, we emphasize that in the present study, the term "extreme" describes values in the tails of the distributions of both considered types of data set rather than record-like events. The limited time span of available satellite measurements results in these extremes also including potentially still relatively moderate seasonal anomalies. However, in the classical peaks-over-threshold sense, it appears reasonable to consider the identified events as extremes."

**Remark 5:**

*"It is also a weak point that by construction every pixel contains the very same amount of "extreme" events. That the heatwave of 2003 did affect only some parts of Europe vanishes from the analysis thereby, and, as the authors rightly point out, for rather stable regions w.r.t. temperature fluctuations and NDVI (think of evergreen forests for example) there is nothing particular or unusual with the values exceeding P90."*

**Answer:**

We understand this concern. However, distinguishing between extreme and less extreme regions would involve the definition of absolute thresholds, above or below which a pixel value may be considered high or low. In turn, our approach intentionally lacks this absolute inter-comparability of values, while at the same time it yields the advantage of considering relatively high or low anomalies for each pixel separately.

**Remark 6:**

*"The distinction of the phenophases is a clear plus for the analysis and, although not unexpected, reveals that the coincidence of the four different combinations of low and high values for LSTAD and NDVI differs very much between the phases. Still, to use the same calendar dates to differentiate the phases is very simplistic and could be improved a lot, given the accumulated knowledge about regional budburst and senescence timings for the different part of Europe."*

**Answer:**

We agree that our differentiation by time periods is very simplistic. Nevertheless, at such a large spatial scale this simplification is almost unavoidable, since the variety of plant communities in Europe is simply too large to consider separately. As for the use of the term "phenophases" for the three considered time periods during the year, we will follow the recommendations of reviewer 1 and taking these plant developmental stages rather as a motivation for defining three distinct periods rather than attributing them consistently to the same phenophases throughout the entire study region. Corresponding changes will be made in our revised manuscript wherever necessary.
* * *
**References in this response letter:**

Coumou, D. and Rahmstorf, S.: A decade of weather extremes, Nature Climate Change, 2, 491–496, doi:10.1038/nclimate1452, 2012.

Dierßen, K.: Vegetation Nordeuropas: 112 Tabellen, Ulmer, Stuttgart, 1996.

Ellenberg, H. and Leuschner, C.: Vegetation Mitteleuropas mit den Alpen: in ökologischer, dynamischer und historischer Sicht, vol. 8104, Ulmer, Stuttgart, 2010

Loew, A. et al.: Validation practices for satellite based earth observation data across communities, Rev. Geophys., 55, doi:10.1002/2017RG000562, 2017.

NOAA: 2011 Seasonal Temperature Anomalies, https://www.climate.gov/news-features/featured-images/2011-seasonal-temperature-anomalies, accessed 2017-08-21, 2012.

Spinoni, J., Lakatos, M., Szentimrey, T., Bihari, Z., Szalai, S., Vogt, J., and Antofie, T.: Heat and cold waves trends in the Carpathian Region from 1961 to 2010, International Journal of Climatology, 35, 4197–4209, doi:10.1002/joc.4279, 2015.

USDA: Spring Dryness and Freeze Lowers Europe's 2007/08 Winter Crop Prospects, https://www.pecad.fas.usda.gov/highlights/2007/05/EU_21May07/, accessed 2017-08-22, 2007.

---

## Author Comment (AC2) · 12 Sep 2017

We are very grateful for the positive overall evaluation and helpful comments raised by the reviewer, which we would like to take as a basis for a thorough revision of our manuscript and encouragement for further work. A detailed point-by-point response, together with answers to the questions raised by the other referees, is provided as a supplementary file accompanying this author comment.

Please also note the supplement to this comment:
https://www.biogeosciences-discuss.net/bg-2017-189/bg-2017-189-AC2-supplement.pdf